# The Effect of Sera from Children with Obstructive Sleep Apnea Syndrome (OSAS) on Human Cardiomyocytes Differentiated from Human Embryonic Stem Cells

**DOI:** 10.3390/ijms222111418

**Published:** 2021-10-22

**Authors:** Hen Haddad, Sharon Etzion, Tatiana Rabinski, Rivka Ofir, Danielle Regev, Yoram Etzion, Jacob Gopas, Aviv Goldbart

**Affiliations:** 1Shraga Segal Department of Microbiology, Immunology and Genetics, Faculty of Health Sciences, Ben Gurion University of the Negev, Beer-Sheva 8410501, Israel; haddadhe@post.bgu.ac.il (H.H.); danirege@post.bgu.ac.il (D.R.); 2Regenerative Medicine & Stem Cell Research Center, Ben-Gurion University of the Negev, Beer-Sheva 8410501, Israel; shar@bgu.ac.il (S.E.); rabinski@bgu.ac.il (T.R.); rivir@bgu.ac.il (R.O.); tzion@bgu.ac.il (Y.E.); 3Cardiac Arrhythmia Research Laboratory, Department of Physiology and Cell Biology, Faculty of Health Sciences, Ben Gurion University of the Negev, Beer-Sheva 8410501, Israel; 4Soroka University Medical Center, Department of Oncology, Ben Gurion University of the Negev, Beer-Sheva 8410501, Israel; 5Soroka University Medical Center, Department of Pediatrics, Faculty of Health Sciences, Ben Gurion University of the Negev, Beer Sheva 8410501, Israel; avivgold@bgu.ac.il; 6Pediatric Pulmonary and Sleep Research Laboratory, Faculty of Health Sciences, Ben Gurion University of the Negev, Beer Sheva 8410501, Israel

**Keywords:** obstructive sleep apnea, sera, inflammation, NF-κB, cardiomyocytes (CM) derived from human embryonic stem cells (hES), intracellular [Ca^2+^]_i_ signaling, beating rate, contractility

## Abstract

Obstructive sleep apnea syndrome (OSAS) patients suffer from cardiovascular morbidity, which is the leading cause of death in this disease. Based on our previous work with transformed cell lines and primary rat cardiomyocytes, we determined that upon incubation with sera from pediatric OSAS patients, the cell’s morphology changes, NF-κB pathway is activated, and their beating rate and viability decreases. These results suggest an important link between OSAS, systemic inflammatory signals and end-organ cardiovascular diseases. In this work, we confirmed and expanded these observations on a new in vitro system of beating human cardiomyocytes (CM) differentiated from human embryonic stem cells (hES). Our results show that incubation with pediatric OSAS sera, in contrast to sera from healthy children, induces over-expression of NF-κB p50 and p65 subunits, marked reduction in CMs beating rate, contraction amplitude and a strong reduction in intracellular calcium signal. The use of human CM cells derived from embryonic stem cells has not been previously reported in OSAS research. The results further support the hypothesis that NF-κB dependent inflammatory pathways play an important role in the evolution of cardiovascular morbidity in OSAS. This study uncovers a new model to investigate molecular and functional aspects of cardiovascular pathology in OSAS.

## 1. Introduction

Obstructive sleep apnea syndrome (OSAS) is a sleep disorder characterized by repetitive nocturnal upper airway obstructive events, associated with intermittent hypoxia [1].

Although OSAS is a frequent disorder in children with a prevalence of 1–5%, pediatric OSAS is under-diagnosed because parents under-report its nocturnal symptoms, and upper airway dysfunction is not always apparent to clinicians [2,3].

Untreated pediatric OSAS can be associated with significant neurobehavioral, cognitive, somatic growth, metabolic, and cardiovascular morbidity, the last considered the leading cause of death in this syndrome in adults [4]. Cardiovascular consequences of OSAS includes: dysregulation of blood pressure (BP), cardiac remodeling, and endothelial dysfunction. Children with OSAS also show decreased cardiac output and oxygen consumption at peak exercise capacity. The most severe pediatric cardiovascular consequence of OSAS is pulmonary hypertension, and Rt, heart failure (Cor pulmonale), if OSAS is untreated. All those conditions are partially reversible with OSAS treatments [5]. In children, surgical treatment, adenotonsillectomy (A and T), is generally considered the first-line therapy for otherwise healthy children who have moderate or severe OSAS and adenotonsillar hypertrophy. Other non-surgical approaches may include the use of anti-inflammatory medications in mild OSAS patients. Current treatments for OSAS are lacking and are accompanied by a variety of complications [6].

In vitro studies showed that intermittent but continuous hypoxia triggers the activation of NF-κB [7]. Furthermore, OSAS induce repeated hypoxia and reoxygenation during sleep, which resembles the condition of ischemia-reperfusion, and generates a large number of reactive oxygen species (ROS). The increased ROS generation triggers the expression of multiple proinflammatory genes via activation of the oxidant-sensitive NF-κB. NF-κB is one of the most important redox responsive transcription factors, and it was shown to play a crucial role in the activation of the promoter activity of over 200 genes, many of which play essential roles in the pathophysiology of atherosclerosis and other cardiovascular diseases [7,8]. Chronic inflammation could lead to cell programmed death, and CM loss due to multiple mechanisms of death is known to occur in cases of pathological cardiac hypertrophy [9,10]. Our laboratory has previously shown that there is a local classical NF-κB overexpression in adenoid and tonsillar tissue of children with OSAS, as well as a systemic activation of the NF-κB by sera of these children [11]. Others also found that NF-κB activity in circulating neutrophils of OSAS patients is increased as compared to control groups. That activity decreases significantly after continuous positive airway pressure (CPAP) treatment. Furthermore, it was found that in neutrophiles isolated from peripheral blood of OSAS patients, the NF-κB activity was substantially higher than in controls [12]. Based on our previous studies on a variety of cell types, including newborn rat cardiomyocytes [11,13], where NF-κB was shown to be activated by sera from OSAS children, we hypothesized that the activation of the NF-κB pathway is involved in the pathophysiology of OSAS-related cardiac morbidity. In this work, we improved our in vitro model by investigating new parameters of beating human cardiomyocytes differentiated from human embryonic stem cells (WA-09 human ES cells) and asked whether incubation with sera from children with OSA increases NF-κB activation and modulates contractile parameters of these cells.

By studying human beating CMs, we have improved our ability to assess by several techniques, the behavior of cells once exposed to disease mimicking conditions such as incubation with sera from OSAS patients. We believe that the use of these cells has significant advantages over transformed cardiomyocytes or primary rodent cells. The findings pave the way to study other conditions related to cardiovascular morbidity in OSAS by this model, such as the biological effects of intermittent hypoxia and monitoring chronic, long-term effects of hypoxia after alleviation of hypoxic conditions.

By better defining and understanding the complex array of molecular, cellular, and physiological factors involved in OSAS, it will be possible to develop short- and long-term controlled anti-inflammatory agents to prevent or ameliorate cardiac morbidity.

## 2. Results

### 2.1. Sera (5%) Is Not Toxic to Cultured Cardiomyocytes

Since all the experiments are based on the comparison between control sera and sera from OSAS children, we first determined the minimal concentration of sera which shows an effect but is not toxic to the hES-derived CM (hESC-CMs) cell line. We incubated the CMs with 5% of OSAS or control sera for 24 h. The XTT assay was used to determine cell viability (Figure 1). The results show that neither control nor OSAS sera are toxic at a 5% serum concentration. This concentration was used thorough this study unless specified otherwise.

### 2.2. NF-κB Is Activated by Stimulation with Sera from OSAS Patients

We have previously demonstrated that NF-κB is activated in the tonsils of OSAS patients, in a SV-40 immortalized human cardiomyocytes cell line as well as in neonatal rat CMs [11,13]. In order to confirm the pro-inflammatory activity of OSAS sera on cells from a target organ in the disease, we incubated hESC-CMs with the sera and determined NF-κB activation. CMs were incubated with 5% sera from OSAS (n = 10) and control children (n = 12) for 2 h (37 °C 5% CO_2_). Following incubation, the active NF-κB subunits p50 and p65 were detected in the nucleus by immunofluorescence. Since only about 40% of the cells were differentiated into cardiomyocytes, (the rest termed as non-cardiomyocytes), the quantitation of fluorescence was determined only on cells stained green with anti-troponin antibody, a known marker for CM. We quantified the intensity of nucleus staining following incubation with several different sera using the Operetta system (Figure 2A). The results show that both p50 and p65 NF-κB subunits are significantly increased in OSAS sera treated hES-CMs. A representative picture is shown in Figure 2B.

### 2.3. Incubation with OSAS Sera Decreases the Beating Rate of CMs

To verify the influence of OSAS sera on cardiomyocyte physiological parameters we tested their beating rate. Initially, we incubated the CMs with 5% sera (similarly to previous experiments), but at this concentration both OSAS and control sera inhibited contraction completely. Therefore, we reduced the serum concentration to 1% and exposed the CMs to OSAS or control sera for 2 h (37 °C 5% CO_2_). The number of beats/minutes was measured thereafter (Figure 3). The results indicate a significant decrease in CMs beating rate after exposing to OSAS sera in comparison to control sera. 

### 2.4. OSAS Sera Affect Intracellular Calcium Signaling and Mechanical Contraction

We examined intracellular calcium [Ca^2+^]_i_ transients and mechanical contraction of hES-CMs grown on a glass coverslip, with the HyperSwitch dual-excitation and dual-emission photometry system and the MyoCam-S fast digital-dimensioning video camera, respectively (IonOptix, MA, USA). All cells showed spontaneous (SP) and synchronized (at 0.5 Hz) [Ca^2+^]_i_ transients and clear contraction responses. Figure 4A shows a decrease in [Ca^2+^]_i_ signaling following 2 min incubation with OSAS sera as compared to control sera. Quantitative analysis of the [Ca^2+^]_i_ transients revealed a decrease in [Ca^2+^]_i_ amplitude (calculated as ΔF/F_0_, Figure 4B) in both OSAS and control sera; however, in the OSAS group, this decrease was significantly larger. The effect of sera on this parameter was reversible after washing the cells with physiological buffer (Figure 4B). The kinetics results, as indicate by the time for [Ca^2+^]_i_ transient decline (calculated by single exponential tau, Figure 4C) and the time for [Ca^2+^]_i_ transient rise (peak time, Figure 4D), were not different between the two groups.

Figure 5 demonstrates the analysis of mechanical contraction. In accordance with the [Ca^2+^]_i_ findings, a reduction in cardiomyocyte contraction amplitude was noted following incubation with OSAS sera (Figure 5B). The half width of the mechanical contraction was unaffected by exposure to the sera (Figure 5C).

## 3. Discussion

Obstructive sleep apnea syndrome is characterized by events of intermittent pauses in breathing during sleep coupled with poor sleep quality [14]. Previous studies on OSAS found evidence of both local (upper airway) and systemic inflammation and also end-organ cardiovascular morbidity [9].

We have previously shown that NF-κB classical pathway monomers p50 and p65 are over expressed compared to controls in adenoids and tonsils that were surgically removed from OSAS patients [11]. We also found elevated activity of NF-κB after OSAS serum incubation in neonatal rat cardiomyocytes and morphological changes on cardiomyocyte transformed cell lines [13]. Encouraged by these findings we further examined cardiomyocytes to assess at the cell level the effects of obstructive sleep apnea syndrome. We demonstrated that NF-κB is activated in rat, mouse and non-beating human immortalized CM, showing evidence of decreased viability and contractility following exposure to OSAS serum. We showed that exposure to patient’s sera is associated with changes in the geometry of the heart at the cellular level [13]. These results contribute to the understanding that there is a major inflammatory pathway that negatively impacts the cardiovascular system of OSAS patients.

In this study, we further developed the idea that the NF-κB pathway may be involved in the pathophysiology of OSAS-related cardiac morbidity. To this end we established a disease in a dish, ex vivo model where human cardiomyocytes with contractile capabilities derived from hES were used. This system offers clear advantages over primary mouse or rat cardiomyocytes or transformed human cardiomyocyte cell lines, which do not contract. Thus, we investigated the effect of sera on cells that represent an affected organ (the heart) in the pathology of the disease.

We first showed that neither sera from OSAS children nor control sera were toxic to the hES derived CMs at 5% concentration for 24 h.

We next found evidence that NF-κB is activated by OSAS sera and expressed in the nucleus of differentiated CM. These results are in line with our previous results with other cell types and with the literature in adult OSAS patients. It is increasingly recognized that OSAS is a systemic rather than a local disorder. The sera of adult patients contain pro-inflammatory cytokines such as IL-6, TNF-α which induces NF-κB mediated inflammation [15,16,17].

Based on these results, we further investigated the influence of OSAS sera on beating rate, calcium homeostasis and mechanical contraction.

Calcium homeostasis is essential for cardiac contractility and normal heart function. Dysregulation in Ca^2+^ handling and its associated proteins may turn into major driving factors towards mechanical and electrical dysfunction and heart failure. It is well known that intracellular calcium release from the sarcoplasmic reticulum (SR) through the RyR2 channel is required for cardiac muscle contraction. During muscle relaxation, calcium is released from the complex actin-myosin-troponin and returns through the SERCA2a pump and phospholamban protein (PLN) back to the SR [18].

In this study, we found a decrease in the beating rate, indicating an inhibitory effect of the OSAS sera on the electrical activity of the CMs. Next, we measured changes in intracellular calcium (ΔF/F0 and calcium kinetics), and mechanical contraction (contraction amplitude, half width) following incubation with different sera. These physiological measurements show a significant impairment in calcium transients following 2 min incubation with OSAS and as a direct effect, a decrease in contraction amplitude. These changes were reversible after washing the cells with physiological buffer, suggesting that protein effectors such as cytokines and/or small molecules in the sera may be involved in inhibition of calcium channels or other components related to calcium induced-calcium release cycle.

Indeed, several studies have demonstrated that cytokines such as TNF-α, IL-1, and IL-6 can influence myocardial function via effects on both myocyte contractility and degradation of extracellular matrix (ECM) [19]. We observed that 5 min incubation of isolated adult cardiomyocytes with IL-6 or TNF-α causes a decline in cell contraction as well as a decrease in peak systolic [Ca^2+^]_i_ and cell shortening amplitude. They suggested that IL-6 exerts a negative inotropic ventricular action through a NO-dependent pathway, while TNF-α produces a negative inotropic action through activation of sphingomyelinase. Prolonged exposure (24 h ± 4) of adult rat cardiomyocytes to IL-1 and IL-6 inhibits cell contractility and [Ca^2+^]_i_ influx. Moreover, a 3 day incubation of neonatal rat cardiac myocytes with IL-1β caused a decrease in both basal contractility (amplitude of contraction, maximum speed of contraction and relaxation) and amplitude of calcium transients. Contractile function and amplitude of calcium transients returned to control values when cells where cultured an additional 3 days in the absence of IL-1-β [20]. Further experiments are needed for understanding the influence of OSAS sera on calcium handling and cardiac function. It is possible that the negative chronotropic and inotropic effects of the OSAS sera that we demonstrate here, lead to chronic compensatory overstimulation of the adrenergic system and the renin-angiotensin-aldosterone system, which are both known to have detrimental consequences on the myocardium [21,22].

### Strengths and Limitations

We have established a novel system to study the effect of OSAS sera and intermittent hypoxic conditions on cardiomyocytes. The use of these cells significantly contributes to the understanding of OSAS and the related cardiovascular pathophysiology. Our results provided evidence that OSAS sera are pro-inflammatory to cardiomyocytes by activating NF-κB. In addition, these sera also affect the physiological behavior of the cells, reducing beating rate, contractibility and modifying Ca^2+^ dynamics.

Our ex vivo model reflects at the cellular level the effects of systemic inflammation. However, this system also has obvious limitations common to other in vitro models, such as the lack of physiological extracellular factors affecting the tissue and the fact that the cardiomyocytes are young cells that do not differentiate to mature cells and are not a complete organ. In addition, the pathology in patients is a result of long term (years) cumulative effects of OSAS. In contrast, here we study the effects in the short time span of hours/days. Therefore, we are careful in our conclusions as pertinent to the disease. Having said that, we believe that the results may open a new window that will enable us to look at the effects of OSAS on cardiomyocytes at the molecular, cellular and physiological levels.

## 4. Materials and Methods

### 4.1. Antibodies

Anti-NF-κB p65 NLS specific [600-401-271]: Rockland Immunochemicals Inc., Limerick, PA, USA.

Anti NFκB p50 (NLS) [sc-114]: Santa Cruz Biotechnology, Dallas, TX, USA.

Rabbit anti NFκB p65 622602: BioLegend, San Diego, CA, USA.

Rabbit anti NFκB p50 [AHP2331]: Bio-Rad Laboratories, Hercules, CA, USA.

Anti-Cardiac Troponin T antibody [1C11], Abcam, Cambridge, UK.

Anti-mouse and anti-rabbit IgG peroxidase, Jackson ImmunoResearch, West Grove, PA, USA.

Goat anti-Mouse IgG (H+L) Highly Cross-Adsorbed Secondary Antibody, Alexa Fluor 488 [A-11029] and Goat anti-Rabbit IgG (H+L) Highly Cross-Adsorbed Secondary Antibody, Alexa Fluor 633 [A-21071], Invitrogen, Thermo Fisher Scientific, Waltham, MA, USA.

Mouse Monoclonal Anti-β-Actin-Peroxidase: Sigma-Aldrich, St. Louis, MO, USA.

Vectashield Mounting Medium with DAPI: Vector Laboratories, Burlingame, CA, USA.

### 4.2. Cell Cultures

Human WA-09 cell line (hES), generated by the WiCell Research Institute.

hES cells were cultured, on Matrigel [23] coated 6-well plates, supplemented with NutriStem hESC XF medium: (Biological Industries, Kibbutz Beit-Haemek, Israel). Cell Dissociation was achieved with Versene Solution (Biological Industries, Kibbutz Beit-Haemek, Israel), at 37 °C for 4 min [24].

### 4.3. Cardiomyocyte (CMs) Differentiation

CM differentiation was performed via modulation of the regulatory elements of Wnt signaling using chemical inhibitors. hES cells were maintained on Matrigel (Corning, NY, USA) in NutriStem medium. Spontaneous cell contraction first appeared on days 7 to 11 of differentiation. The cells were kept in 5% CO_2_ atmosphere at 37 °C heated incubator. Cells were transferred to 96 wells (50 × 10^3^ cells/well) for evaluation of different parameters [24,25].

### 4.4. Patients

This study was approved by the IRB/Helsinki Committee (approval number 0024-SOR, approved: 10 January 2018) and took place in the Department of Pediatrics at the Soroka University Medical Center and the Department of Immunology, Microbiology, and Genetics, Faculty of Health Sciences, Ben-Gurion University of the Negev, Beer-Sheva, Israel. Twenty-four children, who were previously diagnosed with OSAS during an over-night PSG test, were recruited. Fifteen other children with no OSAS were recruited for the control group. The control children underwent tonsillectomy and adenoidectomy (T and A) due to recurrent throat infections and were otherwise healthy at the time. All participants’ parents signed an informed consent form.

Blood samples were collected from the children in the operating room right before undergoing T and A during the morning hours and then centrifuged for 5 min at 1200 rpm. Serum fractions were collected and stored at −80 °C until estimation. Sera from this pool were used in this work. The antropometric and polysomnographic main characteristics of the patients is summarized in our recent publication [13]. The sera from the control and OSAS children were collected at different times. Experiments were performed with as many sera as possible to provide statistically significant results. Several sera were obtained in small quantities were depleted and were not available for all the experiments. Nevertheless, most sera which were obtained in relatively large volumes were common to all the experiments.

### 4.5. XTT-Viability Assay

Differentiated CMs were incubated with 5% control or OSAS sera for 24 h, followed by tetrazolium salt (XTT) solution for 2 h incubation at 37 °C. Metabolically active cells reduce the salt with mitochondrial enzymes to form the orange-colored formazan, which is measured at absorbance wavelengths 475 and 630 nm with a spectrophotometer. Color intensity is proportional to the number of metabolically active cells, hence proportional to cell viability.

### 4.6. Determination of CM Contraction

Contracting CM cells in 96 wells were recorded for later scoring, then they were incubated at 37 °C with 1% control or OSAS sera for 2 h. Immediately after incubation, the same contracting cells in different areas of the well were recorded again. In average, three contracting groups of cells/well (triplicate wells) and beats/minute were determined (an average of nine contracting cells were scored per treatment).

### 4.7. Operetta High-Content Imaging System, PerkinElmer (Akron OH, USA)

Immunofluorescence was used to quantify the activated NF-κB in the nucleus of cardiomyocytes (troponin positive cells). 50,000 differentiated CM cells were plated per well of a 96-well plate and allowed to attach for 24 h in 37 °C. Cells were starved with RPMI 1640 media without FCS for 5 h, after that, the cells were incubated with 5% (non-toxic concentration) of sera from different control or OSAS children for 2 h. Then the cells were washed once with 3% FCS in PBS before being fixed with 4% paraformaldehyde in PBS at room temperature for 20 min. Cells were then washed twice in PBS then twice in 3% FCS in PBS before permeabilization with 0.1% Triton-X100 in PBS for 60 min at room temperature. Cells were washed twice with 3% FBS in PBS before staining with troponin, a cardiomyocyte marker, and NF-κB subunits (P65, P50) in 3% FBS and 0.1% Triton-X100 in PBS for overnight. The cells were then stained with a secondary antibody AF488 (green-troponin) and AF 688 (pink-p65 or p50) for overnight and DAPI (blue-nuclear) staining for 30 min, followed by 3× washing with PBS. Cells were imaged with an Operetta high-content imaging system at 40× magnification. Analysis of the pictures was done through the Columbus server of the company where parameters can be defined such as exclusive fluorescence quantitative scoring of cardiomyocytes (green cells), and the antigen of interest in the nucleus or the cytoplasm (pink) of these cells and not in others (non-cardiomyocytes differentiated cells) in the culture. Four replicate wells were tested for each antibody. Each well is divided into 180 microscope fields (4 × 180 fields) and the pink fluorescence of only green cells (cardiomyocytes) is scored automatically. Quantitative immunohistochemistry, Operetta (Appendix A).

### 4.8. Measurements of Intracellular Calcium Transients and Mechanical Contraction

Embryonic stem cells were differentiated to CMs on a glass coverslip coated with Matrigel and analyzed at days 15–19 of differentiation.

### 4.9. Intracellular Calcium

Intracellular calcium [Ca^2+^]_i_ signals were measured with the HyperSwitch dual-excitation and dual-emission photometry system (IonOptix, Westwood, MA, USA). Differentiated hES-CMs were incubated for 10 min at 37 °C with medium containing 5-μM Indo-1AM (Molecular Probes) and Pluronic F-127 (Life Technologies) at a dilution of 1:1, followed by a 10-min wash with B-27 supplement medium to remove excess dye. The cells were transferred to a perfusion chamber mounted on the stage of an inverted microscope and perfused with HEPES-buffered Tyrode’s solution (TB) containing (in mmol/L) NaCl 140, KCl 5.4, MgCl_2_ 1, sodium pyruvate 2, CaCl2 1, HEPES 10, and glucose 10 (pH 7.4) at room temperature [26]. The cells were stabilized with perfused TB, followed by 4–5 min with no perfusion, and then the cells were exposed to different 5% sera (control or OSAS). The signals were recorded before exposure to sera, immediately and 2 min after sera supplement, followed by perfusion with TB and wash for 5 min. After 5 min, [Ca^2+^]_i_ was recorded, and the experiment was complete. Signals from 2–3 sera supplements were assessed on each slide with 10 min wash between experiments. [Ca^2+^]_i_ transients were measured by the F405/F480 ratio of Indo-1AM fluorescence. Signals were recorded during stimulation at 0.5–0.75 Hz using a field stimulator (MyoPacer; IonOptix, Westwood, MA, USA) through two platinum electrodes placed on the sides of the perfusion chamber. Data were analyzed using an IonWizard data acquisition system (IonOptix). Parameters were calculated from an average of 15 to 20 successive transients under each condition. [Ca^+2^]_i_ transient amplitude was expressed as the delta amplitude from peak to baseline. Control sera were evaluated in triplicate and five OSAS sera were evaluated in duplicate.

### 4.10. Mechanical Contraction

Mechanical contraction was evaluated using MuscleMotion, a powerful and versatile ImageJ macro to measure in vitro or in vivo muscle contraction or motion [27]. CMs contraction was continuously recorded using a MyoCam-S, a fast digital dimensioning video camera (IonOptix, Westwood, MA, USA) connected to Virtual Dub software 1.10 4, (ISkySoft, Hertfordshire, UK).

Cells were exposed to different 5% sera (control or OSAS). Parameters were calculated from the 30-s movie for each condition. Contraction amplitude was expressed as the difference between contraction signals during maximal systolic contraction and diastolic relaxation, half-width duration was measured from 50% raise to 50% decline of the contraction transients.

### 4.11. Statistical Analysis

Values are expressed as mean ± SEM. Statistical analysis was performed using Prism 8.0 (GraphPad Software, San Diego, CA, USA). Comparisons between control and OSA sera groups were performed using unpaired Student’s *t*-test. In cases in which n was lower than 6, Mann–Whitney test was performed instead. For comparison of physiological measures in the different time points (Figure 4 and Figure 5) data were analyzed by one-way ANOVA with Holm-Sidak’s multiple comparison post-test. In cases in which n was lower than 6, Kruskal–Wallis with Dunn’s multiple comparison post-test was performed instead. The specific tests that were used are mentioned in the legend of each figure. The criterion for significance was set at *p* < 0.05. Unless otherwise stated *p*-values are displayed graphically as follows: * *p* < 0.05, ** *p* < 0.01, *** *p* < 0.001.

## Figures and Tables

**Figure 1 ijms-22-11418-f001:**
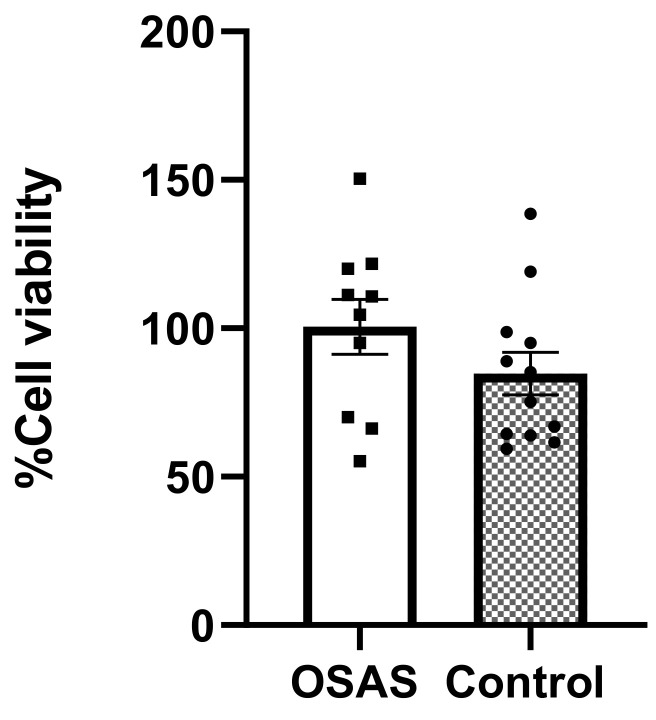
Incubation of cells with 5% OSAS or control sera is not toxic to cells. Sera were tested for toxicity on hES-CMs cells. 50,000 cells/well were plated in triplicate of 96-well plate and allowed to attach for 24 h at 37 °C. Next the cells were incubated with OSAS (n = 12) or control (n = 10) sera for 24 h followed by incubation with the XTT solution for 2 h. Absorption was determined at 475, 630 nm. Squares and circles represent individual sera. Percentage of cell viability was determined in comparison to untreated cells, the difference was not significant (*t*-test *p* = 0.1868).

**Figure 2 ijms-22-11418-f002:**
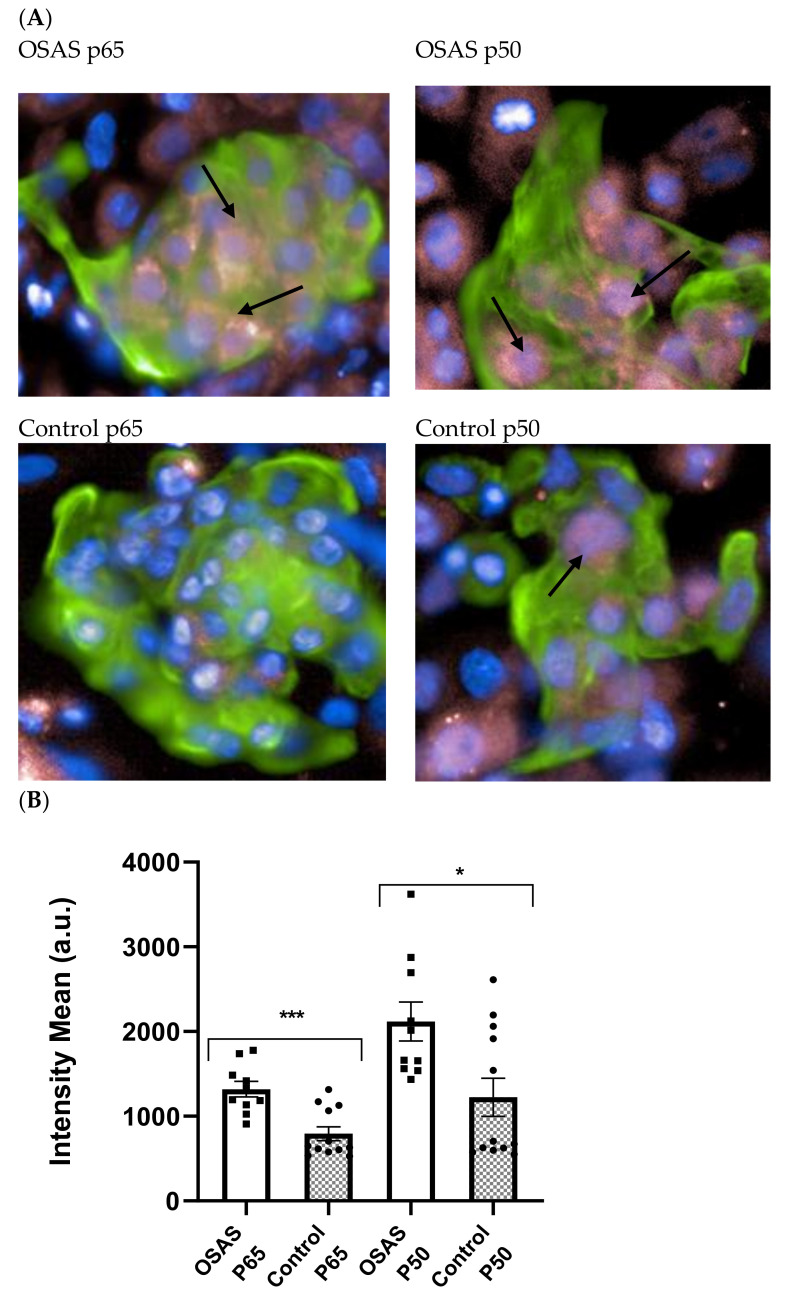
NF-κB is activated by stimulation with sera from OSAS patients. (**A**) A representative picture of a cluster of cells is presented. Cardiomyocytes are identified by staining with anti-cardiac troponin (green). NF-κB subunits: Anti p50 or anti p65 (pink). Arrows point to nuclei expressing p50 or p65. (**B**) The total average intensity of NF-κB subunits p50 and p65 staining was measured in the nuclei of cardiomyocytes. The cells were incubated each time with 12 different OSAS sera and compared to 10 different control sera (5%). The results presented are an average of 3 separate experiments done in duplicates, showing a significant increase of nuclear p50 and p65 following incubation with OSAS sera. Squares and circles represent individual sera. Statistical significance was determined by the Student’s *t*-test comparing cells incubated with control or OSAS sera (p50 *p* = 0.01, p65 *p* = 0.001). * *p* < 0.05, *** *p* < 0.01.

**Figure 3 ijms-22-11418-f003:**
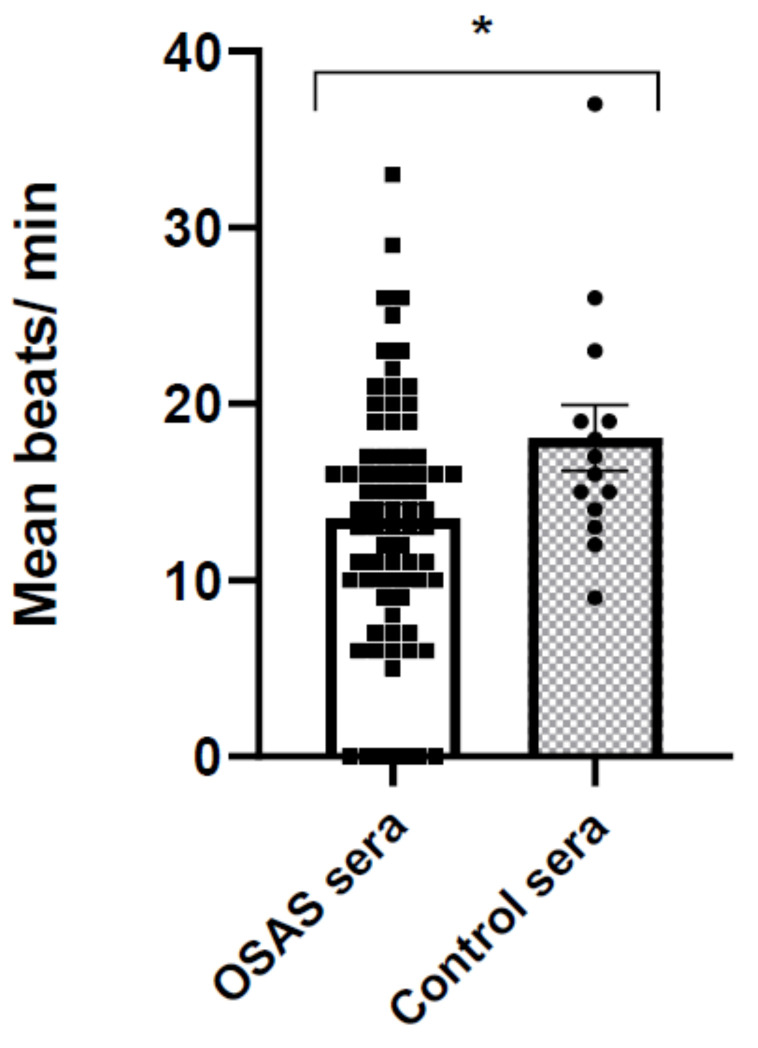
Incubation with OSAS sera decreases the beating rate of CMs. CMs were incubated for 2 h with 1% OSAS (n = 24) or control sera (n = 14). The areas where cells contracted before and after incubation with sera were recorded and scored (contractions/minute). Each serum was scored in triplicate wells. Squares and circles represent individual sera. In each well, 3–4 beating groups of cells were scored, total of 9–12 groups with the same treatment were scored. Statistical significance of *p* = 0.0274 was determined by the Student’s *t*-test. * *p* < 0.05.

**Figure 4 ijms-22-11418-f004:**
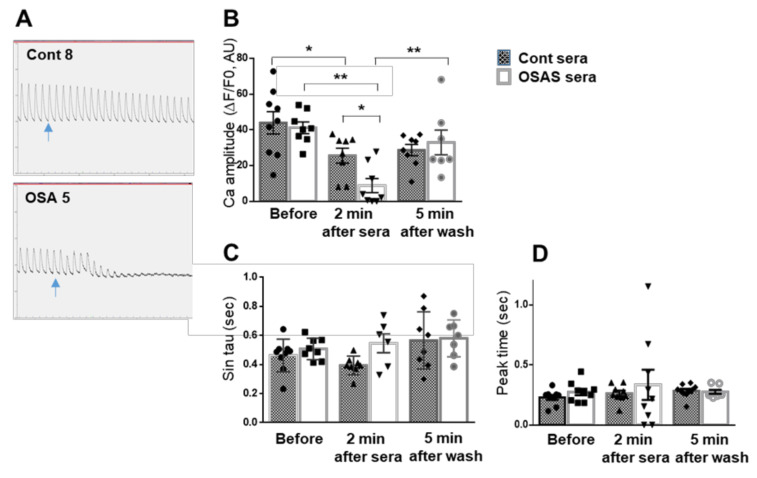
OSAS sera reduce intracellular calcium signaling. (**A**) An example of [Ca^2+^]_i_ transients upon field stimulation at 0.5Hz before and immediately after the addition of control or OSAS sera (5%) (blue arrow). The signals were recorded before exposure to sera, and two minutes after sera supplement, followed by perfusion with TB and wash for 5 min. Then [Ca^2+^]_i_ was recorded, and the experiment was completed. Several physiological parameters were measured as shown in figures (**B**–**D**). (**B**) Ca^2+^ amplitude was calculated as ∆F/F0, meaning the change that occurred in [Ca^2+^]_i_ signaling during the contraction–relaxation cycle. The results demonstrate that following control sera (n = 9) there is a significant decrease in Ca^2+^ amplitude (about 40%, *p* = 0.039) while washing the cells with physiological buffer did not alter the Ca^2+^ amplitude (*p* = 0.9). Following OSAS (n = 8), on the other hand, there is about 75% decrease in Ca^2+^ amplitude (*p* < 0.001) and a recovery of the transient after washing the cells (*p* = 0.008). As shown in B, there is a significant difference between the Ca^2+^ amplitudes 2 min after adding control (n = 9) as compared to OSAS (n = 8) sera (*p* = 0.011). Following 5 min wash with physiological buffer both groups show the same Ca^2+^ amplitude (*p* = 0.9). (**C**) (sin tau) demonstrates the time for re-uptake of the Ca^2+^. (**D**) (Peak time) demonstrates the time for max Ca^2+^ peak. Data display Mean ± SE. Unpaired Student’s *t*-test was used to compare between control and OSAS sera within each period. Squares, triangles, diamonds and circles represent individual sera. One way ANOVA was used to analyze the differences between the different periods (before, 2 min after sera, 5 min after wash) in Control or OSAS sera. * *p* < 0.05, ** *p* < 0.01.

**Figure 5 ijms-22-11418-f005:**
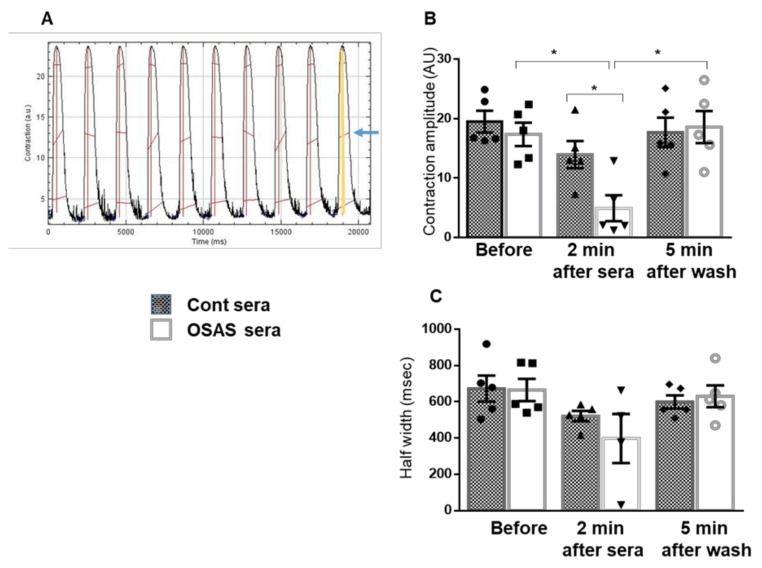
OSAS sera reduce cardiomyocytes mechanical contraction. (**A**) An example of mechanical contraction traces and analysis upon field stimulation at 0.5 Hz. Short movies (30 s) were recorded before exposure to sera and 2 min after sera supplement (control, n = 5 or OSAS, n = 5), followed by perfusion with TB and wash for 5 min. (**B**) Contraction amplitude was calculated as delta of peak high from baseline. (**C**) Half-width duration was calculated as delta of the time at 50% transient raise to the time at 50% decline of the contraction transients. The results show a significant decrease (*p* = 0.040) in contraction amplitude 2 min after exposure to OSAS sera. The amplitude returned to baseline after wash (*p* = 0.027). As shown in B, there is a significant difference between the contraction amplitudes 2 min after adding control (n = 5) as compared to OSAS (n = 5) sera (*p* = 0.016). There is no change in the contraction kinetics as described by the half-width duration. Data display as Mean ± SE. The non-parametric Mann–Whitney test was used to compare between control and OSAS sera within each period. Squares, triangles, diamonds and circles represent individual sera. Kruskal-Wallis test was used to analyze the differences between the different periods (before, 2 min after sera, 5 min after wash) in control or OSAS sera. * *p* < 0.05.

## Data Availability

Deidentified individual participant data will not be made available.

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
