# Peer review of "The Effect of Sera from Children with Obstructive Sleep Apnea Syndrome (OSAS) on Human Cardiomyocytes Differentiated from Human Embryonic Stem Cells"

_ijms, 2021, doi:10.3390/ijms222111418_

Round 1
Reviewer 1 Report
I have read with an interest this original paper of Henet al. The authors took up an interesting topic of the effect of sera from children with OSAS on human cardiomyocytes differentiated from human embryonic stem cells. In my opinion, the following issues need correction:
Major:
1) The introduction should be concluded with the hypothesis of the study.
2) The Statistical Analysis section remains practically undescribed. It must be corrected.
3) Institutional Review Board Statement: the number of the proper decision must be added!
Minor:
1) The following statements required proper references:
a) "Obstructive Sleep Apnea Syndrome (OSAS) is a sleep disorder characterized by repetitive nocturnal upper airway obstructive events, associated with intermittent hypoxia."
b) Untreated pediatric OSAS can be associated with significant neurobehavioral, cognitive, somatic growth, metabolic, and cardiovascular morbidity, the last considered the leading cause of death in this syndrome in adults.
2) You stated that "Obstructive Sleep Apnea Syndrome (OSAS) is a sleep disorder characterized by repetitive nocturnal upper airway obstructive events, associated with intermittent hypoxia.". It is worth underlying that the level of hypoxia indicators - e.g. HIF-1alpha - remains chronically upregulated as it was shown: 10.5664/jcsm.8682. Interestingly this effect is pronounced even after one-night CPAP therapy.
3) Please provide a separate "Strengths and limitations" section at the end of the paper.
4) "In this study, we found a decrease in beating rate (Fig 3) indicating an inhibitory effect of the OSAS sera on the pacemaker electrical activity of the CMs." - the meaning of strikethrough (of the word "pacemaker") should be provided.
5) Figures:
a)1: Add information about lack of significant differences (e.g. N.S.), would you?
b) 2b/3/4: please provide p-values on the picture
Author Response
We thank the reviewer for the comments. This is our point by point response.
Reviewer 1
Major:
1) The introduction should be concluded with the hypothesis of the study.
Indeed, we stated our hypothesis in the the introduction and expanded on its significance:
….we hypothesized that the activation of the NF-κB pathway is involved in the pathophysiology of OSAS-related cardiac morbidity…
2) The Statistical Analysis section remains practically undescribed. It must be corrected.
The statistics were fully described for each type of experiments and in the materials and methods section
3) Institutional Review Board Statement: the number of the proper decision must be added!
Added in the text
Minor:
1) The following statements required proper references:
a) "Obstructive Sleep Apnea Syndrome (OSAS) is a sleep disorder characterized by repetitive nocturnal upper airway obstructive events, associated with intermittent hypoxia."
b) Untreated pediatric OSAS can be associated with significant neurobehavioral, cognitive, somatic growth, metabolic, and cardiovascular morbidity, the last considered the leading cause of death in this syndrome in adults.
Appropriate references were added (Refs 1 and 4 respectively)
2) You stated that "Obstructive Sleep Apnea Syndrome (OSAS) is a sleep disorder characterized by repetitive nocturnal upper airway obstructive events, associated with intermittent hypoxia.". It is worth underlying that the level of hypoxia indicators - e.g. HIF-1alpha - remains chronically upregulated as it was shown: 10.5664/jcsm.8682. Interestingly this effect is pronounced even after one-night CPAP therapy. The reference was added in the text (Ref 2)
3) Please provide a separate "Strengths and limitations" section at the end of the paper.
Added at the end of the discussion
4) "In this study, we found a decrease in beating rate (Fig 3) indicating an inhibitory effect of the OSAS sera on the pacemaker electrical activity of the CMs." - the meaning of strikethrough (of the word "pacemaker") should be provided.
The strikethrough line on “pacemaker” was erased.
5) Figures:
a)1: Add information about lack of significant differences (e.g. N.S.), would you? Lack of significant differences relates to p values above 0.05 as indicated in the text and legends to figures.
b) 2b/3/4: please provide p-values on the picture.
Significant * were added on the figures and explained in the legends
Reviewer 2 Report
The study shows that OSAS sera are pro-inflammatory to cardiomyocytes by activating NF-κB. These sera also affect the physiological behavior of the cells, reducing beating rate, contractibility and modifying Ca2+ dynamics.
General comments
The study appears to have appropriate methodology, however the data presentation in Figures should be corrected (lack of information considering the statistical significances – described in specific points), as well as captations (see specific points). Authors use in different experiments various numbers of sera, for example in point 2.2 - sera from OSAS (n=10) and control children (n=12), wheras in point 2.3 OSAS (n=24) and control sera (n=14). What is more, this information is missing in point 2.4. The information explaining this approach (using different number of sera in various experiments) should be added. It is described that OSAS sera has proinflammatory activity, but no measurement of any proinflammatory cytokine in sera was made. I would suggest to measure proinflammatory cytokines panel in analyzed sera using one of the multiplex assay kits, allowing to simultaneously detect the levels of multiple cytokines in a single sample. This way the authors would be able to explain more precisely observed OSAS sera effects. Authors should not refer to the Figures in the Discussion section, I would recommend to combine Results with Discussion section.
Specific points
Page 2
In the middle of second paragraph - CPAP abbreviation is not explained, does the authors mean continuous positive airway pressure (CPAP) therapy?
In the middle of second paragraph - in the cited paper [11] - there are neutrophils, not monocytes,
in the same cited paper [11] - elevated neutrophil NF-κB activity not p65 NF-κB subunit amount
Page 3
Figure 1 - instead of giving value of p in the captation, please add star and appropiate line (between two columns) directly to the figure, in the same time please explain what star (*) means in the captation.
Page 4
Figure 2 – there is inconsistence between captation and figure, panel A is described as B; arrows described in the captation do not exist on the presented cells photos; instead of giving value of p in the captation, please add stars and appropiate lines (between adequate two columns) directly to the figure, in the same time please explain what stars (* and **) mean in the captation; please add % concentration of sera and time of the incubation.
Page 5
Paragraph 2 – please add number of sera used in described experiment.
Figure 3 - please add time of the incubation; instead of giving value of p in the captation, please add star and appropiate line (between two columns) directly to the figure, in the same time please explain what star (*) means in the captation; please add % concentration of sera and time of the incubation.
Page 6
First paragraph – please add % concentration of sera and number of sera used in experiment.
Second paragraph – second sentence must be corrected (was noted is used twice for the same thing); please add % concentration of sera
Figure 4 - please add % concentration of sera, time of the incubation, number of sera used in experiment; in panel C - instead of giving value of p in the captation, please add star and appropiate line (between two columns) directly to the figure, in the same time please explain what star (*) means in the captation; panel C – it repeats informations from panel B, the units in Y axis are incorrect, for % should be between 0-100 %, I would suggest removing this panel, relevant description in the main text is sufficient.
Page 7
Figure 5 – panels B and C are not described in the captation; no explanation to star (*) presented on the figure (panel B); numbers of analyzed sera (n) are not given.
Discussion
Second paragraph – lack of reference for the following sentence: ‘We also found elevated activity of NF-κB after OSAS serum in-cubation in human transformed cell lines’.
Page 8 – definitely it should be used TNF-α, instead of TNF-a or TNF 1 α.
In the Discussion the authors must not refer to the Figures, I would recommend to combine Results with Discussion section.
Author Response
Reviewer 2.
The study appears to have appropriate methodology, however the data presentation in Figures should be corrected (lack of information considering the statistical significances – described in specific points), as well as captations (see specific points).
The statistical significances between columns have been introduced in the figures (*) and explained in the legends as detailed in the specific points section below. Furthermore we expand the statistical analysis section in the material and methods.
Authors use in different experiments various numbers of sera, for example in point 2.2 - sera from OSAS (n=10) and control children (n=12), wheras in point 2.3 OSAS (n=24) and control sera (n=14). What is more, this information is missing in point 2.4. The information explaining this approach (using different number of sera in various experiments) should be added.
The sera from control and OSAS children were collected at different times. Experiments were done with as many sera as possible to provide statistically significant results. Several sera were obtained in small quantities were depleted and were not available for all the experiments. Nevertheless, most sera which were obtained in relatively large volumes were common to all the experiments.
We have added this paragraph to the Materials and Methods section.
The number of sera used in point 2.4 was added to the figure legend
It is described that OSAS sera has proinflammatory activity, but no measurement of any proinflammatory cytokine in sera was made. I would suggest to measure proinflammatory cytokines panel in analyzed sera using one of the multiplex assay kits, allowing to simultaneously detect the levels of multiple cytokines in a single sample. This way the authors would be able to explain more precisely observed OSAS sera effects.
The reviewer’s point is well taken, especially since proinflammatory cytokines in sera from OSAS patients have already been described by others and by us in tonsils of pediatric patients (Ref.12). We decided that the measurement of cytokines is not in the scope of this paper since many samples would needed to be tested, each with a wide variety of different cytokine expression patterns. We foresee that the results would be difficult to interpret. Instead, we have preliminary data for a near-future paper in which we have already identified specific cytokines produced by the same cardiomyocytes cell line used here, following a 12 hour cyclic hypoxia protocol, mimicking a real life situation.
Authors should not refer to the Figures in the Discussion section, I would recommend to combine Results with Discussion section.
Specific points
Page 2
In the middle of second paragraph - CPAP abbreviation is not explained, does the authors mean continuous positive airway pressure (CPAP) therapy? Explained in the text
In the middle of second paragraph - in the cited paper [11] - there are neutrophils, not monocytes, Corrected
in the same cited paper [11] - elevated neutrophil NF-κB activity not p65 NF-κB subunit amount. Corrected
Page 3
Figure 1 - instead of giving value of p in the captation, please add star and appropiate line (between two columns) directly to the figure, in the same time please explain what star (*) means in the captation. Corrected
Page 4
Figure 2 – there is inconsistence between captation and figure, panel A is described as B; arrows described in the captation do not exist on the presented cells photos; instead of giving value of p in the captation, please add stars and appropiate lines (between adequate two columns) directly to the figure, in the same time please explain what stars (* and **) mean in the captation; please add % concentration of sera and time of the incubation.
The figure has been corrected
Page 5
Paragraph 2 – please add number of sera used in described experiment. Added
Figure 3 - please add time of the incubation; instead of giving value of p in the captation, please add star and appropiate line (between two columns) directly to the figure, in the same time please explain what star (*) means in the captation; please add % concentration of sera and time of the incubation. Information added and the figure corrected
Page 6
First paragraph – please add % concentration of sera and number of sera used in experiment. Added
Second paragraph – second sentence must be corrected (was noted is used twice for the same thing); please add % concentration of sera Corrected
Figure 4 - please add % concentration of sera, time of the incubation, number of sera used in experiment; in panel C - instead of giving value of p in the captation, please add star and appropiate line (between two columns) directly to the figure, in the same time please explain what star (*) means in the captation; panel C – it repeats informations from panel B, the units in Y axis are incorrect, for % should be between 0-100 %, I would suggest removing this panel, relevant description in the main text is sufficient.
The figure was corrected as suggested and panel C removed
Page 7
Figure 5 – panels B and C are not described in the captation; no explanation to star (*) presented on the figure (panel B); numbers of analyzed sera (n) are not given.
Figure 5 was corrected accordingly
Discussion
Second paragraph – lack of reference for the following sentence: ‘We also found elevated activity of NF-κB after OSAS serum in-cubation in human transformed cell lines’.
The sentence was corrected and a reference added
Page 8 – definitely it should be used TNF-α, instead of TNF-a or TNF 1 α.
Corrected
In the Discussion the authors must not refer to the Figures, I would recommend to combine Results with Discussion section.
We eliminated in the discussion the reference to the figures but kept the Discussion separate from the Results. We leave it to the editor’s discretion
Round 2
Reviewer 1 Report
The authors addressed all the comments sufficiently improving the manuscript.